# A Comparison in Patient Comfort Using Conventional Syringe and Needleless Jet Anesthesia Technique in Periodontal Surgery—A Split-Mouth Randomized Clinical Trial

**DOI:** 10.3390/medicina58020278

**Published:** 2022-02-12

**Authors:** Preethi Shankar, Burnice Nalina Kumari Chellathurai, S. Ashok Kumar, Jaideep Mahendra, Maryam H. Mugri, Mohammed Sayed, Mohammad Almagbol, Mohammed Hussain Dafer Al Wadei, Rajaram Vijayalakshmi, Namasivayam Ambalavanan, A. Thirumal Raj, Shankargouda Patil

**Affiliations:** 1Department of Periodontology, Faculty of Dentistry, Meenakshi Ammal Dental College and Hospital, Alapakkam Main Road, Chennai 600095, India; preethishankar95@hotmail.com (P.S.); drashok.perio@madch.edu.in (S.A.K.); drjaideep.perio@madch.edu.in (J.M.); drvijaya.perio@madch.edu.in (R.V.); profhod.perio@madch.edu.in (N.A.); 2Department of Maxillofacial Surgery and Diagnostic Sciences, College of Dentistry, Jazan University, Jazan 45412, Saudi Arabia; dr.mugri@gmail.com; 3Department of Prosthetic Dental Sciences, College of Dentistry, Jazan University, Jazan 45412, Saudi Arabia; drsayed203@gmail.com; 4Department of Community and Periodontics, Faculty of Dentistry, King Khalid University, Abha 61421, Saudi Arabia; malmagbol@kku.edu.sa; 5Department of Restorative Dental Science, College of Dentistry, King Khalid University, Abha 61421, Saudi Arabia; moalwadai@kku.edu.sa; 6Department of Oral Pathology and Microbiology, Sri Venkateswara Dental College and Hospital, Chennai 600130, India; thirumalraj666@gmail.com; 7Department of Maxillofacial Surgery and Diagnostic Sciences, Division of Oral Pathology, College of Dentistry, Jazan University, Jazan 45142, Saudi Arabia

**Keywords:** jet injections, local anesthesia, periodontal debridement

## Abstract

*Background and Objectives:* Periodontal surgery requires local anesthetic coverage to alleviate patient discomfort. Needles and injections can engender feelings of fear and anxiety in individuals. This study aimed to assess the level of comfort and anxiety in patients during the administration of local anesthesia using needleless jet anesthesia (JA) when compared to a conventional syringe (CS) in periodontal surgery. *Method and Materials:* 60 sites were designated for injection in a split-mouth design in 30 subjects who required periodontal surgery. Local anesthesia was administered in two appointments scheduled one week apart using either a JA system or a CS. The Visual Analogue Scale (VAS), Verbal Rating Scale (VRS), and Beck’s anxiety inventory were used to report the pain and anxiety levels while injecting local anesthesia. Statistical analysis of the results was performed using the Shapiro–Wilks test and Paired t-test. *Results:* Patients reported greater comfort with JA. The VAS and VRS values were statistically significant—(*p* = 0.003) and (*p* = 0.001), respectively. Patients showed fear and were nervous about receiving a local anesthetic using a CS. A few subjects experienced lingering pain with the CS, whereas greater comfort and no lingering soreness were reported post-operatively at the site of JA administration. *Conclusions:* This study provides the first comprehensive assessment of using JA for periodontal surgical procedures. Lower pain scores were consistently observed with the use of jet injectors. Patients were at ease and reported lesser anxiety and greater comfort with jet injectors, making it ideally suited for providing local anesthesia in periodontal surgery.

## 1. Introduction

Periodontal diseases are a range of inflammatory diseases caused due to bacterial buildup [1]. If left untreated, these diseases can lead to the deterioration of tooth-supporting structures and eventually tooth loss [2]. Long-term studies show that removing plaque both supragingivally and subgingivally promotes healing and prevents inflammation and periodontal tissue deterioration [3]. Depending on the severity of the disease, periodontal treatment can be restricted to non-surgical or could involve surgical intervention. Both treatments could be an unpleasant experience for the patient and hamper the patient’s day-to-day activity [4]. An aversion to pain combined with fear and anxiety about dental treatment are the most common causes for a patient’s hesitation to undergo routine dental care. “Needle phobia” or “blenophobia” is the fear of needles and it affects one in five adults [5].

From the patient’s perspective, injection of local anesthetics with a conventional needle syringe is unpleasant and is the source of fear and anxiety. Injections are a crucial factor for premature termination or delay of treatment. There is a documented link between anxiety and fear of pain, as well as the actual feeling of pain. Anxiety and dread leads to stress, which lowers a patient’s pain threshold [6]. Needles and anesthetic agents have undergone an evolution in quality and design over the last few decades. However, the method of anesthetic administration remains unchanged. A traditional needle anesthetic delivery causes pain at the puncture and injection stages [7]. The pain is caused by improper syringe handling, compounded by excessive pressure on the plunger and rapid injection of large amounts of anesthetic fluid [8,9].

Literature reveals several approaches to reduce the uncomfortable sensation during local anesthesia such as administering topical anesthetics before injection [10], employing a computerized injection system [11], manually adjusting the injection speed [12], and using needleless jet injection devices.

Needle-free jet injections are an easy, painless mode of anesthetic delivery [13]. Their design allows for injection of a minimal volume of anesthesia when compared to the conventional technique A spring is connected to an apparatus in a needleless system which applies sufficient pressure to the ampoule’s plunger, allowing the anesthetic solution to travel through a micro orifice at a high speed [14]. The absence of a needle in a jet injection provides a comfortable pain-free experience for the patient. It eliminates the puncture and sluggish injection phases, both of which can be painfully uncomfortable [15].

Previous studies compared the needleless jet technique to other anesthetic techniques in nonsurgical periodontal treatments revealed that the jet technique has superior efficacy [9]. The needleless jet technique can provide long-lasting anesthesia with lesser pain, and more comfort, thus creating a positive environment for further treatment.

The present study was designed as a randomized split-mouth clinical trial aimed to compare and evaluate the level of comfort and anxiety in patients during the administration of local anesthesia using needleless jet anesthesia (JA) when compared to a conventional syringe (CS) in periodontal surgery. The null hypothesis of the study was that the patients’ pain experience was similar in both techniques. The alternate hypothesis was that patients experienced lesser pain in the needleless jet anesthesia technique when compared to the conventional syringe technique.

## 2. Materials and Methods

### 2.1. Ethical Approval and Clinical Trial Registry

This study was approved by the Institutional review board of Meenakshi Ammal Dental College, Chennai-600095 (MADC/IRB-XXXIII/2020/552), India. The study was conducted following the Helsinki declaration and registered with the clinical trial registration (CTRI/2020/10/028527). The 2010 CONSORT guidelines were followed for this RCT. Written informed consent was obtained from all participants in this study.

Patient population: 30 subjects with bilateral periodontal disease or mucogingival deformities who required periodontal surgery were registered after obtaining informed consent. 60 sites were designated for injection.

Study design: A single-blinded, randomized control trial with a split-mouth design was used, and the treatment quadrants for each patient were assigned at random using a coin toss procedure.

Test site (Group A): 30 sites with periodontal disease or mucogingival deformities which required periodontal surgery were administered with local anesthesia using JA (0.5 mL).

Control site (Group B): 30 sites with periodontal disease or mucogingival deformities which required periodontal surgery were administered with local anesthesia using a CS (2 mL).

Sample size calculation and power of the study: In this investigation, α was set to 0.05, with a maximum of 52 sites accepting 5% and a study power of 95%. To show a meaningful difference in the primary outcome of this RCT, a sample size of 60 sites in thirty patients in each group was required.

Inclusion criteria:Participants are in the age group of 18 to 50 years;Systemically healthy male/female patients;Subjects requiring bi-lateral periodontal surgery.

Exclusion criteria:Participants who are allergic to local amide anesthetics;Pregnant and lactating women;Alcoholics and smokers;Patients who used analgesics or medication acting on the central nervous system.

Procedure:

Patients with bilateral surgical intervention (I/IV quadrant and II/III quadrant) in a split-mouth design were scheduled for two appointments one week apart. In the first appointment, a jet injection was administered at the test site (Figure 1). The local anesthetic agent used was lidocaine 2% with epinephrine 1:80,000. The jet injector has a needleless syringe that can hold up to 5 mL of local anesthetic solution, as well as a body with a discharge button and an adapter CAP-01 (anterior/posterior sectors) or the adapter CAP-02 (palate), which can be altered for each patient and ensures pinpoint precision at the injection site. The needleless syringe was filled with 0.5 mL of 2% lidocaine from the local anesthetic vial with an adapter head. After loading and placing the needleless syringe, the adapter CAP01 was placed at 90° on the attached gingiva and shot for the infusion (Figure 1).

Patients were recalled after one week for the second appointment for treatment at the control site. The control site was injected with local anesthesia using the CS. The CS was loaded with a 2 mL local anesthetic agent (lidocaine 2% with epinephrine 1:80,000) and injected using the buccal infiltration technique.

A questionnaire was handed over before and after the local anesthetic injections and experiences both at the test site and the control site. Visual Analogue Scale (VAS) and Verbal Rating Scale (VRS) were used for pain assessment at each appointment. VAS used a 100-mm scale was used to assess pain. The left endpoint was labeled “no pain” while the right endpoint was labeled “worst suffering imaginable [16]. VRS is a five-point scale that includes no, mild, moderate, severe, and extremely severe pain [17].

Beck Anxiety Inventory:

The Beck’s anxiety scale is a self-reporting 21-questionnaire format (Figure 2). The patients were asked to fill out this questionnaire and give a score from 0 to 3 for each of the 21 common symptoms of anxiety. The level of anxiety on using either technique was noted using this anxiety scale [18]. The patients were asked to fill out the forms before commencing the procedures. Figure 3 depicts the entire study design.

### 2.2. Statistical Analysis

The statistical analysis was performed using SPSS (IBM SPSS Statistics for Windows, Version 21.0, Armonk, NY, USA: IBM Corp., 2020) software. A parametric test (Shapiro–Wilks test) was used to compare data between groups and non-parametric tests (paired t-test) were used to analyze the Visual Analogue Scale (VAS), Verbal Rating Scale (VRS), and Beck’s anxiety scale values, respectively. The significance level was fixed as 5% (α = 0.05).

## 3. Results

The Visual Analogue Scale (VAS) scale for pain was recorded from 0 (no pain) to 100 (extreme pain). The values were lesser for the JA group reported lower scores for pain (52.066 ± 13.617) compared to the CS group. The JA scores depicted mild discomfort. The results were statistically significant (*p =* 0.003).

The Verbal Rating Scale (VRS) scale for pain was lower in the JA group compared to the CS group. Patients reported moderate pain in the JA group, which was lesser than with the CS, which was highly significant (*p* = 0.001).

Both the VAS and the VRS values indicate that the discomfort felt by the JA group was lower than that experienced by the CS group. The higher scores in the CS groups indicate greater discomfort and are statistically significant (*p* = 0.003). A verbal rating score of 4 was found for CS compared to JA which had a score of 2. This indicates the patient’s feeling of extreme pain with CS. Table 1 depicts the Mean Visual Analogue Score (VAS) for the CS group and JA groups.

The anxiety scale values revealed that patients were terrified (*p* = 0.007), nervous (*p* = 0.018), and scared (*p* = 0.002) of CS injection compared to JA. The results were statistically significant. Patients were terrified, or afraid, nervous, and scared of the CS technique far more than the JA. Table 2 depicts the anxiety scores according to Beck’s anxiety Inventory.

## 4. Discussion

Local anesthesia has evolved through various compositions and different modes of administration. However, administration through traditional CS remains the mainstay across the globe. Needleless jet injectors are a newer model of injection of local anesthesia in periodontal surgery. They have seen successfully used over the years in the field of medicine. The findings of our study suggest that jet injectors are an effective alternative to CS techniques and can be successfully used for periodontal surgery. The pain experienced was lower with the JA than the CS and the associated anxiety was also lower with JA.

Studies reveal that drugs administered through jet injectors show outstanding bioavailability. Nevertheless, jet injectors have a few drawbacks. Complaints of occasional bruising, pain, and discomfort have held back the wide acceptance of jet injectors [19]. The patient’s experience of pain during anesthetic administration fosters a negative attitude towards dental treatment. Minimizing this pain component can engender wider acceptance of necessary dental care. Compared to the CS technique, JA demonstrated better patient acceptance, comfort, and compliance for pressure anesthesia (70%) than with CS anesthesia (20%) [20].

The effectiveness of the local anesthetic solution deposited through puncturing the mucosa using a CS depends on several factors. The amount of solution and the pressure with which it is injected need to be considered. The jet injection works on the principle of spring recoil action and pressure [21], depositing the anesthesia in a small fraction of time resulting in a higher diffusion rate. Thus, a higher volume of anesthetic solution can be deposited. The recoil spring mechanism in the jet injector is credited for the higher diffusion of anesthetic solution within the tissues.

The anesthetic gets diluted progressively as it diffuses through the nerve due to the presence of fluids in the extracellular tissues. It is also absorbed by the non-neural tissues, capillaries, and lymphatic vessels in the infiltrated area. As long as the local anesthetic solution is present in the nerve, the nerve impulses that signal pain to the brain are blocked. This interval is termed the duration of anesthesia [7].

The onset of action and latency period are key factors to be considered in the adoption of JA. The onset of anesthesia can vary depending on the anesthetic substance and how different anesthetic procedures are modified [22,23]. The onset of action for infiltration using the CS technique with 2% lignocaine solution is 2 min. A study comparing the onset of action in maxillary buccal infiltration using 4% articaine and 2% lignocaine found that it was at 1.6 min and 2.8 min, respectively [22]. JA has an onset of action of less than thirty seconds for most patients. Their onset of action is rapid compared with the CS technique. A study by Oliveira AC et al., 2019 found the latency period for JA to be two minutes to achieve pulpal anesthesia to conduct root canal treatments [14].

In this study, patients preferred the JA to the CS anesthesia and reported feeling better anesthetic effect with jet injectors. A study conducted by Rajan Gupta et al. in 2018 stated that patients preferred Eutectic Mixture of Local Anesthesia (EMLA) application over JA as the application of EMLA was less traumatic and had desired anesthetic properties. Patients appear wary of jet anesthetic because of its bulky appearance [24].

Konstantinos Nikolaos et al. compared children’s acceptance and preference between the jet injector and conventional injection technique. Counterintuitively, most children (73.6%) preferred the traditional method to the needleless jet injector [25]. This may be in no small part to the unconventional appearance of the jet injector that may have disconcerted the children. Chetana et al. examined the acceptance, preference, and efficacy of jet injectors against traditional anesthetic techniques in adult patients undergoing dental restorative treatments. They reported that adults preferred the use of jet injectors citing greater comfort and lesser pain and fear with jet injectors [26]. There were no reports of lingering pain at the site of injection with jet injectors. The soreness at the site of injection is a common post-operative complaint with the conventional injection technique.

Tissue toxicity is not a concern when local anesthetics are placed topically on mucous membranes due to the relatively short application duration, fast absorption into the circulation, and strong regenerating capability of the oral mucosa. With every shot of the jet injector, the anesthetic solution is delivered at a depth of 2 to 2.5 mm below the epithelium, and one-tenth of an ml solution is deposited. With the velocity and distance of penetration, the periodontal apparatus (gingiva, periodontal ligament, and bone) was anesthetized making it hassle-free to perform periodontal surgeries.

All forms of periodontal therapy—scaling and root planing, periodontal flap treatment, frenectomy, vestibuloplasty, and root coverage procedures—were carried out using the needleless jet and conventional injection technique. Many of the participants who had a fear of dental treatment requiring administration of local anesthesia with needle and syringe reported that the needleless jet anesthetic technique was more comfortable and would opt for the same for future treatments. Subjects were assessed with Beck’s anxiety inventory and reported that they were more comfortable and felt less terrified of the needleless jet injector.

There are a few limitations to JA when compared to the CS. The gun-like shot sound from the JA might exacerbate the anxious patients who are already uncomfortable and afraid at the prospect of an injection. The sound is caused by the recoil spring-like action as the solution is deposited at high pressure. The device might be bulky to hold and operate even though it can be carried around handily. The size and appearance of a JA injector are larger compared to the CS. The sight of a large injector could be apprehensive to patients and alarm them, expanding feelings of dread.

Despite these shortcomings, they can be valuable in patients who have needle phobia. With sufficient explanation and counseling to acclimatize the patient, jet injectors could be used widely to treat anxious patients. It eliminates the hassle of sharp needle disposal and accidental punctures in the dental clinic. Unlike other needle-free techniques such as sprays, pills, patches, etc., jet injectors can be used with existing drug preparations and commercially available compositions which are designed for conventional needle-based injections. This brings down the inventory requirements and is cost-saving. The cartridges are sterilized using a disinfecting agent and the CAP is discarded after every use. Jet injectors can serve in various surgical procedures such as periodontal flap therapy, periodontal esthetic procedures, extraction, and root canal treatment. They can be used to deliver various dosages based on the procedure and the intensity of anesthesia required. They are also user-friendly. Jet injectors do not require any specialized training to handle the instrument professionally.

## 5. Conclusions

This paper is the first comprehensive assessment of the feasibility of using JA for periodontal surgery. The jet injection technique was widely accepted by patients due to the reduced pain on injection and they were less anxious and afraid of receiving an anesthetic injection with the jet injector. Further studies are required to assess the latency time. Clinically, JA provides reliable durable local anesthetic coverage with fewer complications. It is user-friendly, cost-effective with a short learning curve for the professional, and is suited for numerous procedures. In summary, based on our findings, we conclude that needleless jet anesthesia offers superior comfort and acceptance to a conventional syringe.

## Figures and Tables

**Figure 1 medicina-58-00278-f001:**
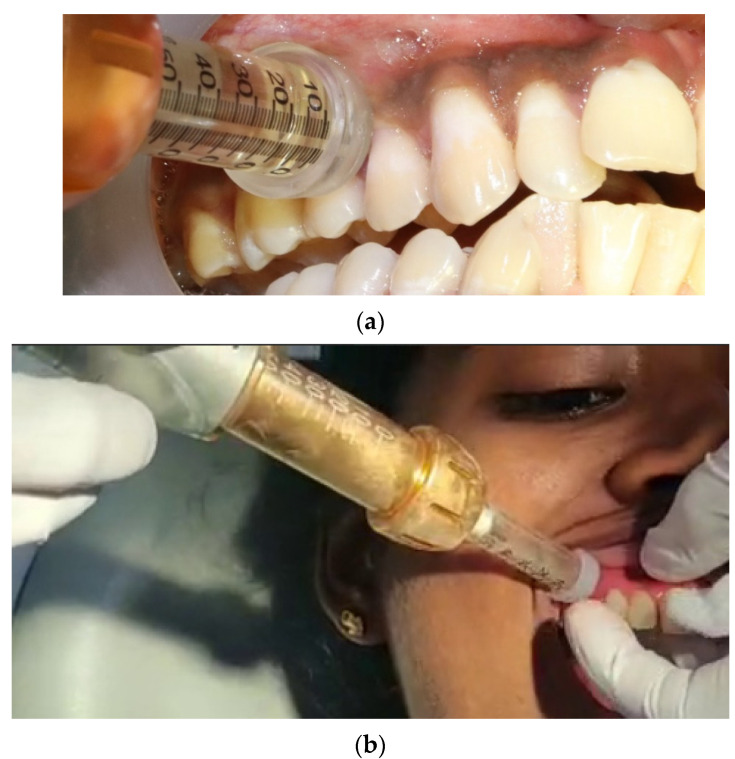
Jet anesthesia (JA) (**a**) Depicts the application of local anesthesia using the needleless. jet injection concerning the 14, 15 tooth region before periodontal therapy. (**b**) The adapter CAP01 is placed at 90° on the attached gingiva and a shot of infusion is achieved.

**Figure 2 medicina-58-00278-f002:**
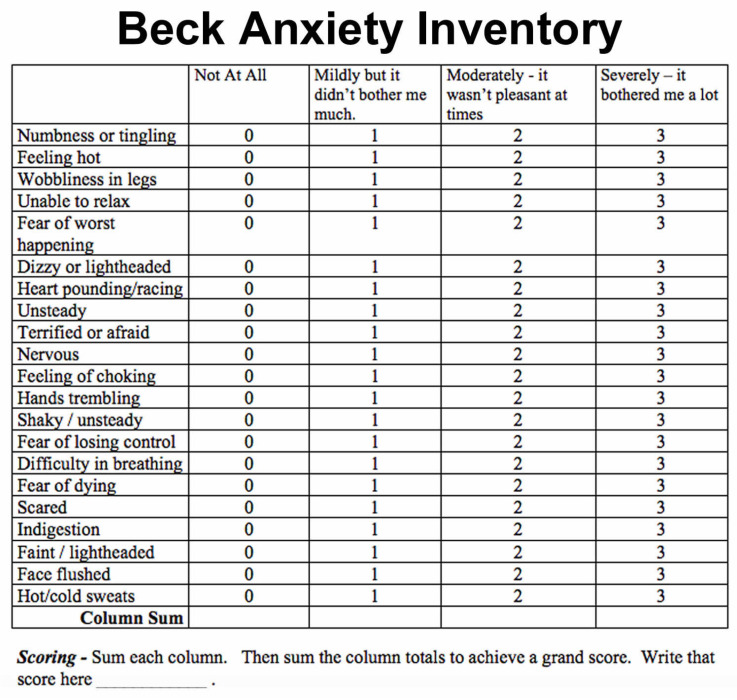
Beck’s anxiety scale in a 21-questionnaire format.

**Figure 3 medicina-58-00278-f003:**
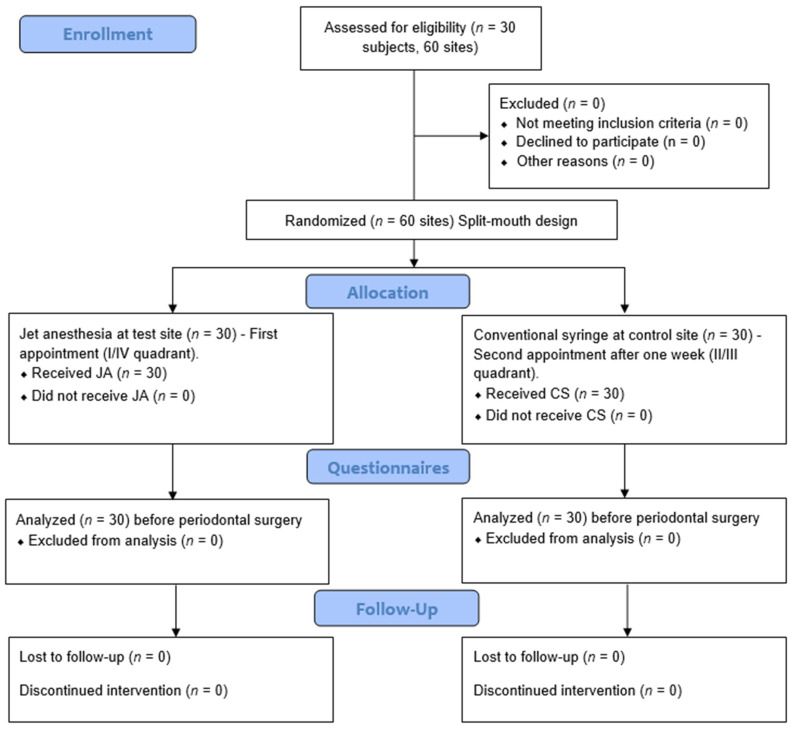
Flowchart of study design.

**Table 1 medicina-58-00278-t001:** Mean Visual Analogue Scores (VAS) for CS and JA group.

	Control Group (CS)	Test Group (JA)	*p*-Value
VAS score (Mean ± SD)	60.233 ± 12.789	52.066 ± 13.617	0.003 *
VRS score (Meadian)	4(2)	2(2.5)	0.001 *

***** Statistically significant (*p* = 0.001).

**Table 2 medicina-58-00278-t002:** Anxiety scores according to Beck’s Anxiety Inventory in CS and JA group.

Anxiety Scale (Median)	Control Group (CS)	Test Group (JA)	*p*-Value
Terrified or Afraid	1(1)	1(1)	0.007 *
Nervous	1(0)	1(1)	0.018 *
Scared	1(1)	1(1)	0.002 *

***** Statistically significant.

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
