# Peer review of "A Comparison in Patient Comfort Using Conventional Syringe and Needleless Jet Anesthesia Technique in Periodontal Surgery—A Split-Mouth Randomized Clinical Trial"

_medicina, 2022, doi:10.3390/medicina58020278_

Round 1

Reviewer 1 Report

This clinical study compared two anesthetic methods when treating periodontal disease. This topic is interesting. However, there are several scientific considerations that should be taken into account.

  1. General comments:
  • Please doublecheck, if you have uploaded the recent version of your manuscript!?
  • Please use the template

 https://www.mdpi.com/files/word-templates/medicina-template.dot

when preparing the manuscript and fill out every section completely and correctly. Also, be aware of subheadings (Abstract: Background and Objectives: //. Materials and Methods: // Results: // Conclusions:) and word counts.

  • Scientific English proofreading is highly recommended. Proof-reading companies may help.
  • Shorter sentences better guide the reader.
  • Fundamental basic references are missing.
  1. Title:
    • You do not test efficacy. What is an effective anesthesia? This is very difficult to explain. Your rather investigate patient acceptance determined by questionnaires!
    • I would suggest the study type at the end
    • Is the study really randomized? As far as I understand you have chosen a split-mouth design.
  2. Abstract
    • Please rewrite using correct subheadings and word count
  3. Key-Words:
    • Please use Mesh-Terms and remove “anesthesia” being double.
  4. Introduction:
    • Please rewrite and restructure. Please, cite fundamental basic references, especially when focusing on periodontal disease.
    • Also, several supplemental parts can be removed (Mucogingival excursus in 1 section)
    • A hypothesis after presenting the aim of the study would be beneficial.
  5. Materials and Methods:
    • Begin with study center and patient population
    • First paragraph: Do you mean a sample size calculation?
    • I suggest including the test and control site paragraphs in a study design and procedures referring the flow-chart.
    • Please be sure including a validated questionnaire and if so, please mention.
    • Include a statistics paragraph
    • Use a comprehensible abbreviation for each test (e.g., JA) and control (e.g., CS) group also in tables and figures
    • Please give some more information concerning the anxiety scale.
    • Mention if the study is in line with the consort statement referring.
    • If I hardly understand correctly, the study is in split-mouth design. This should be clarified in the flow-text, title and flow-chart to better guide the reader.
    • “Patients with bilateral periodontal disease / mucogingival deformities is a too vague. The inclusion spectrum is too broad and cannot be compared. I would suggest including scaling and root-planing left and right mouth-side only.
  6. Results:
    • Please mention key-results and avoid to describe a table of figure.
    • Present pure results without any interpretation
    • When using VAS: Please explain what stands for e.g., extreme pain (100) and no pain (0). “A right and left endpoint” is rather confusing.
  7. Figures:
    • Be in line with author guidelines, inserting figures in the text flow promptly after referring.
    • In figure 1 you do not see anything concerning the technique and in combination with the subheading rather confuses the reader. It can be removed and replaced by a figure much more educational.
    • Figure 2 can be removed as it explains a standard dental procedure.
    • The flow-chart should be handled like a figure. I suggest using a self-made flow-chart with important information in bold letters.
  8. Tables:
    • Use table templates (see above)
    • Be in line with author guidelines, inserting tables in the text flow promptly after referring.
    • Explain abbreviations in the subheading
    • Table 1 totally confuses the reader. He is not able to follow (abbreviations!). Keep a table and its subheading explainable without reading the whole manuscript.
    • See above, without additional information regarding the anxiety scale, the reader is unable to understand the table
  9. Discussion:
    • Please rewrite and structure
      • Start with the key-findings (the first paragraphs are rather introduction).
      • What are strengths and limitations?
    • Discuss strengths and limitations particularly comparing results in relation to other studies
    • What are and future directions?
  • Conclusions:
    • Please rewrite and focus on the conclusions for clinicians.
    • The whole paragraph is really confusing.
    • The first part can be removed as it is rather introduction.
    • In the previous parts of your manuscript is not obvious that patients scheduled for periodontal surgery are included. Different surgical methods are too broadly chosen (see below). I would suggest including scaling and root planing.
  • References:
    • Fundamental basics are missing

Author Response

Reviewer 1:

QUERY

CLARIFICATION

This clinical study compared two anesthetic methods when treating periodontal disease. This topic is interesting. However, there are several scientific considerations that should be taken into account.

  1. General comments:
  • Please doublecheck, if you have uploaded the recent version of your manuscript!?
  • Please use the template

·       The recent version of the manuscript template has been used to make the necessary changes.

When preparing the manuscript and fill out every section completely and correctly. Also, be aware of subheadings (Abstract: Background and Objectives: //. Materials and Methods: // Results: // Conclusions:) and word counts.

  • Scientific English proofreading is highly recommended. Proof-reading companies may help.
  • Shorter sentences better guide the reader.
  • Fundamental basic references are missing.

·       The necessary headings have been added to the abstract and the word limit has been adhered to in the revised version.

·       English proof reading has been done and necessary changes have been made.

  1. Title:
    • You do not test efficacy. What is an effective anesthesia? This is very difficult to explain. Your rather investigate patient acceptance determined by questionnaires!
    • I would suggest the study type at the end
    • Is the study really randomized? As far as I understand you have chosen a split-mouth design.

·       The title has been altered keeping in mind the goal of the study and orienting it towards patient comfort rather than efficacy.

·       Study type is added to the title

·       Yes, it is a split mouth randomized study

3.     Abstract

    • Please rewrite using correct subheadings and word count

·       The abstract has been shortened to the word limit and the subheading have been altered.

·        

4.     Key-Words:

    • Please use Mesh-Terms and remove “anesthesia” being double.

·       The proper Mesh terms have been checked for and altered.

5.     Introduction:

    • Please rewrite and restructure. Please, cite fundamental basic references, especially when focusing on periodontal disease.
    • Also, several supplemental parts can be removed (Mucogingival excursus in 1 section)
    • A hypothesis after presenting the aim of the study would be beneficial.

·       The fundamental studies have been cross checked and the required references have been added and quoted. Context that seemed irrelevant have been removed or altered.

·       The hypothesis of the study has been mentioned at the end of the introduction.

6.     Materials and Methods:

    • Begin with study center and patient population
    • First paragraph: Do you mean a sample size calculation?
    • I suggest including the test and control site paragraphs in a study design and procedures referring the flow-chart.
    • Please be sure including a validated questionnaire and if so, please mention.
    • Include a statistics paragraph
    • Use a comprehensible abbreviation for each test (e.g., JA) and control (e.g., CS) group also in tables and figures
    • Please give some more information concerning the anxiety scale.
    • Mention if the study is in line with the consort statement referring.
    • If I hardly understand correctly, the study is in split-mouth design. This should be clarified in the flow-text, title and flow-chart to better guide the reader.
    • “Patients with bilateral periodontal disease / mucogingival deformities is a too vague. The inclusion spectrum is too broad and cannot be compared. I would suggest including scaling and root-planing left and right mouth-side only.

·       The order of the content has been changed accordingly.

·       The sample size calculation was mentioned in the first paragraph.

·       The Beck’s anxiety questionnaire has been referred to and the the questionnaire and scoring has been added to the text.

·       Necessary short forms have been addressed in the paragraphs.

·       CONSORT 2010 guidelines have been mentioned and the necessary flowchart is added.

·       This study is a split mouth design study and the same has been added to in the title as well for better clarity.

·       The speciality of this study is to include periodontal surgical cases alongside nonsurgical therapy as such kind of a study has never been done before. Hence I have mentioned as bilateral periodontal disease and mucogingival surgeries.

  1. Results:
    • Please mention key-results and avoid to describe a table of figure.
    • Present pure results without any interpretation
    • When using VAS: Please explain what stands for e.g., extreme pain (100) and no pain (0). “A right and left endpoint” is rather confusing.

·       Key results have been discussed in the revised version and the meaning of scores and the appropriate abbreviations are mentioned.

  1. Figures:
    • Be in line with author guidelines, inserting figures in the text flow promptly after referring.
    • In figure 1 you do not see anything concerning the technique and in combination with the subheading rather confuses the reader. It can be removed and replaced by a figure much more educational.
    • Figure 2 can be removed as it explains a standard dental procedure.
    • The flow-chart should be handled like a figure. I suggest using a self-made flow-chart with important information in bold letters.

·       The figure one has been altered and an another image of the device completely has been added.

·       Figure two has been removed

·       The necessary descrption has been altered.

·       The flowchart has been added as a picture.

  1. Tables:
    • Use table templates (see above)
    • Be in line with author guidelines, inserting tables in the text flow promptly after referring.
    • Explain abbreviations in the subheading
    • Table 1 totally confuses the reader. He is not able to follow (abbreviations!). Keep a table and its subheading explainable without reading the whole manuscript.
    • See above, without additional information regarding the anxiety scale, the reader is unable to understand the table

·       The required template has been adhered to.

·       The abbreviations are explained.

·       The table is split into two for better understanding with proper headings.

·       The decimal places have been kep to three uniformly for all values.

  1. Discussion:
    • Please rewrite and structure
      • Start with the key-findings (the first paragraphs are rather introduction).
      • What are strengths and limitations?
    • Discuss strengths and limitations particularly comparing results in relation to other studies
    • What are and future directions?

·       The discussion has been altered in the necessary manner.

·       The limitations have been added to a new paragraph and highlighted.

·       The strengths were already mentioned in the last paragraph of the discussion.

  1. Conclusions:
    • Please rewrite and focus on the conclusions for clinicians.
    • The whole paragraph is really confusing.
    • The first part can be removed as it is rather introduction.
    • In the previous parts of your manuscript is not obvious that patients scheduled for periodontal surgery are included. Different surgical methods are too broadly chosen (see below). I would suggest including scaling and root planing.

·       The conclusions has been rewritten and put accordingly.

·       As mentioned above, the various periodontal therapies are mentioned in this study and not only scaling and root planing as the novelty of this study lies in the inclusion of periodontal surgical therapy.

  1. References:
    • Fundamental basics are missing

·       The references have been changed accordingly.

Reviewer 2:

The manuscript is very well written, uses up-to-date and relevant literature for the scientific field. Some doubts arose in the review, and therefore I ask the authors for clarification/suggestion of corrections.

QUERY

CLARIFICATION

1.     The flowchart presented on page 6 seems unnecessary to me. As there were no follow-up losses nor the need to analyze the losses, I would only insert this information in the text;

·       The flowchart has been converted as an image and added in the text where it belongs. The flowchart is in lines with the 2010 CONSORT and is mentioned for the same purpose as it is a randomized clinical trial- split mouth design.

2.     Doubts in table 1 - especially in the specific topics of the anxiety scale - the medians presented do not indicate statistical significance in the independent items (subtopics of the scale).... it is necessary to review the results! It is also necessary to standardize the number of decimal places in the table.

·       The number of decimal places are standardized to three for all the values.

·       The results were reviewed and only three out of the eighteen questionnaire format had positive and significant clinical significance as statistically calculated.

3.     An item seems to me to be very important to be presented in the discussion - what are the limits of the proposed new method? Cost? Is other equipment needed? Does it require differentiated professional training? Even though these elements were not analyzed in the work, these possible limits/possibilities of the technique are expected to be presented to the reader.

·       The limitations and merits of the study have been described in the last two paragraphs of the discussion. Required additions of missing points have been made.

I congratulate the authors for their excellent work, waiting promptly for the necessary clarifications.

Reply: The authors thank the reviewer for their positive comments

Reviewer 2 Report

The manuscript is very well written, uses up-to-date and relevant literature for the scientific field. Some doubts arose in the review, and therefore I ask the authors for clarification/suggestion of corrections:
- the flowchart presented on page 6 seems unnecessary to me. As there were no follow-up losses nor the need to analyze the losses, I would only insert this information in the text;
- doubts in table 1 - especially in the specific topics of the anxiety scale - the medians presented do not indicate statistical significance in the independent items (subtopics of the scale).... it is necessary to review the results! It is also necessary to standardize the number of decimal places in the table.
- an item seems to me to be very important to be presented in the discussion - what are the limits of the proposed new method? Cost? Is other equipment needed? Does it require differentiated professional training? Even though these elements were not analyzed in the work, these possible limits/possibilities of the technique are expected to be presented to the reader.
I congratulate the authors for their excellent work, waiting promptly for the necessary clarifications.

Author Response

(The authors gave the same response as above.)

Round 2

Reviewer 1 Report

The authors have revised the manuscript, but some of my previous remarks have not been taken into account. Unfortunately, some remarks should still be altered before publication.

  1. Abstract:
    1. The abstract can be shortened in some parts (background, conclusion) to gain space for the type of surgery, time-span between interventions and timepoint VAS being filled out.
    2. Be aware using the indication (=periodontal surgery) and kind of. (Conventional and plastic surgery)
  2. Introduction:
    1. You mean supra- and subgingival plaque removal!
    2. According the guideline treating periodontal disease, you first start with non-surgical debridement. The severity of the disease is not decisive.
    3. Is the spring used to control injection pressure or the volume of the anesthetic?
    4. The study was designed as a randomized split-mouth clinical trial.
    5. Again, you do not evaluate efficacy! See my previous review!
  3. Materials and Methods
    1. The power-calculation was performed on side-level (52) and not on patient level!
    2. Be aware using JS and CS within the whole manuscript.
    3. Please retract the inclusion and exclusion columns.
    4. The patients are scheduled for a bi-lateral surgical intervention (I/IV quadrant and II/III quadrant) (split-mouth design)!
    5. Be aware using the indication under in- and exclusion criteria and procedure (=periodontal surgery). Mixing periodontal therapy and periodontal surgery is rather confusing.
    6. Please give some more information regarding the “Beck Anxiety Inventory” in ONE subtitle. You use two which are alone and in combination uninterpretable (one in the screenshot of your table regarding a scoring: What is scored? And one very general. What is the reason of this “Inventory”?
    7. When were the questionnaires handed out to the patients, please include?
    8. I would suggest to include the first paragraph (page 5, line 149-151) in figure 1`s subtitle.
  1. Results:
    1. The key-results are presented well. Refer to the tables afterwards.
    2. The 5th paragraph can be removed as a conclusion it is rather discussion.
    3. When was the questionnaire filled out by participants? The handing out (before and after intervention) is not of interest
  2. Discussion:
    1. Please add the key findings of your study in the first section (see my previous review!)
    2. This section can be shortened as information appears in duplicate either in the introduction or in the discussion section itself.
    3. As far as I see, the timepoint when the questionnaire being filled can be a limitation. Also, including conventional periodontal surgery (open flap debridement) and plastic surgery is a limitation due to the fact, that the recoalescence (postoperative pain) of latter being much more intense.
    4. What about sterilization and reprocessing of the Jet anesthesia?
  3. Conclusion:
    1. Please shorten!
  4. Flow-Chart:
    1. Include the treatment/anesthetic methods. “Allocated control intervention” is too vague.
    2. As far as I see, the participants are not scheduled for a follow-up visit as the questionnaires were directly handed out to the patients.
    3. Please mention that patients were scheduled for the second quadrant after one week and include the timepoint of participant being asked using questionnaires!
    4. By “Allocated” you mean anesthesia? And by “Analyzed” you mean questionnaires handed out to the participants? What is the time-span?
    5. Be aware of correct English (analyzed!) and blank spaces.

Author Response

Reviewer 1:

QUERY

CLARIFICATION

1.    Abstract:

1.    The abstract can be shortened in some parts (background, conclusion) to gain space for the type of surgery, time-span between interventions and timepoint VAS being filled out.

2.    Be aware using the indication (=periodontal surgery) and kind of. (Conventional and plastic surgery)

·      The abstract has been shortened and proper uniform terminologies have been made use of in all the context.

2.    Introduction:

1.    You mean supra- and subgingival plaque removal!

2.    According the guideline treating periodontal disease, you first start with non-surgical debridement. The severity of the disease is not decisive.

3.    Is the spring used to control injection pressure or the volume of the anesthetic?

4.    The study was designed as a randomized split-mouth clinical trial.

5.    Again, you do not evaluate efficacy! See my previous review!

·      Yes, it is changed to supra and subgingival.

·      Yes, surgical therapy is determined only after SRP.

·      The spring only controls the pressure and not the volume.

·      The necessary changes have been made to the manuscript.

3.    Materials and Methods

1.    The power-calculation was performed on side-level (52) and not on patient level!

2.    Be aware using JS and CS within the whole manuscript.

3.    Please retract the inclusion and exclusion columns.

4.    The patients are scheduled for a bi-lateral surgical intervention (I/IV quadrant and II/III quadrant) (split-mouth design)!

5.    Be aware using the indication under in- and exclusion criteria and procedure (=periodontal surgery). Mixing periodontal therapy and periodontal surgery is rather confusing.

1.    Please give some more information regarding the “Beck Anxiety Inventory” in ONE subtitle. You use two which are alone and in combination uninterpretable (one in the screenshot of your table regarding a scoring: What is scored? And one very general. What is the reason of this “Inventory”?

6.    When were the questionnaires handed out to the patients, please include?

7.    I would suggest to include the first paragraph (page 5, line 149-151) in figure 1`s subtitle.

·      Yes, the power calculation was done site specific.

·      The abbreviations JA and CS have been mentioned appropriately in the revised manuscript.

·      The inclusion and exclusion criteria could not be retracted as it forms the base for he kind of patients to be included for the study and it is patient oriented.

·      The specification for bilateral surgery and split mouth design have been added to the revised manuscript.

·      Appropriate terminologies have been mentioned- periodontal surgery.

·      A paragraph of the anxiety scale has been added.

·      Questionnaires were handed out before performing the periodontal surgery when the the two techniques were going to be performed.

·      The subtitle for figure 1 is typed accordingly.

4.    Results:

1.    The key-results are presented well. Refer to the tables afterwards.

2.    The 5th paragraph can be removed as a conclusion it is rather discussion.

3.    When was the questionnaire filled out by participants? The handing out (before and after intervention) is not of interest

·      The tables are referred to in the latter.

·      5th paragraph has been removed

·      Before the intervention the anxiety was assessed.

5.    Discussion:

1.    Please add the key findings of your study in the first section (see my previous review!)

2.    This section can be shortened as information appears in duplicate either in the introduction or in the discussion section itself.

3.    As far as I see, the timepoint when the questionnaire being filled can be a limitation. Also, including conventional periodontal surgery (open flap debridement) and plastic surgery is a limitation due to the fact, that the recoalescence (postoperative pain) of latter being much more intense.

4.    What about sterilization and reprocessing of the Jet anesthesia?

·      The key findings have been added to the first paragraph.

·      The repeated points have been removed or altered.

·      The limitation of questionnaire doesn’t apply as it is only regarding the pain on GIVING anesthesia using the two techniques and not the pain after the surgical intervention.

·      A line on the disinfection of the jest anesthesia is mentioned.

6.    Conclusion:

1.    Please shorten!

.

·      The conclusion has been shortened.

7.    Flow-Chart:

1.    Include the treatment/anesthetic methods. “Allocated control intervention” is too vague.

2.    As far as I see, the participants are not scheduled for a follow-up visit as the questionnaires were directly handed out to the patients.

3.    Please mention that patients were scheduled for the second quadrant after one week and include the timepoint of participant being asked using questionnaires!

4.    By “Allocated” you mean anesthesia? And by “Analyzed” you mean questionnaires handed out to the participants? What is the time-span?

5.    Be aware of correct English (analyzed!) and blank spaces

·      The allocated intervention is mentioned in the revised manuscript.

·      Follow up regarding lingering pain at the site of anesthesia was assessed.

·      Time span is mentioned.

·      The spellings and blank spaces are corrected.
